# Microsyntenic Clusters Reveal Conservation of lncRNAs in Chordates Despite Absence of Sequence Conservation

**DOI:** 10.3390/biology8030061

**Published:** 2019-08-24

**Authors:** Carlos Herrera-Úbeda, Marta Marín-Barba, Enrique Navas-Pérez, Jan Gravemeyer, Beatriz Albuixech-Crespo, Grant N. Wheeler, Jordi Garcia-Fernàndez

**Affiliations:** 1Department of Genetics, Microbiology and Statistics, Faculty of Biology, University of Barcelona, 08028 Barcelona, Spain; 2School of Biological Sciences, University of East Anglia, Norwich Research Park, Norwich NR4 7TU, UK; 3German Cancer Research Center, 69120 Heidelberg, Germany

**Keywords:** lncRNAs, genome_evolution, synteny, amphioxus

## Abstract

Homologous long non-coding RNAs (lncRNAs) are elusive to identify by sequence similarity due to their fast-evolutionary rate. Here we develop LincOFinder, a pipeline that finds conserved intergenic lncRNAs (lincRNAs) between distant related species by means of microsynteny analyses. Using this tool, we have identified 16 bona fide homologous lincRNAs between the amphioxus and human genomes. We characterized and compared in amphioxus and *Xenopus* the expression domain of one of them, *Hotairm1*, located in the anterior part of the Hox cluster. In addition, we analyzed the function of this lincRNA in *Xenopus*, showing that its disruption produces a severe headless phenotype, most probably by interfering with the regulation of the Hox cluster. Our results strongly suggest that this lincRNA has probably been regulating the Hox cluster since the early origin of chordates. Our work pioneers the use of syntenic searches to identify non-coding genes over long evolutionary distances and helps to further understand lncRNA evolution.

## 1. Introduction

Identifying and understanding the factors that underlie the evolution of morphological complexity is one of the central issues in the field of evolutionary developmental biology (or evo-devo). From the initial claims that gene duplication and neofunctionalization were at the core of phenotypic change [1], the current view also takes into account the fine-tuning of gene regulation [2] and increasing the proteome and interactome complexity through additional processes. In this regard, molecular mechanisms such as alternative splicing or RNA-editing, and the RNA world, with molecules like small miRNA or long non coding RNAs (lncRNAs), allow deeper and multifaceted levels of gene regulation [3]. LncRNA-mediated regulation stands out as a quick and efficient mechanism to modulate gene expression, as these molecules are function-ready almost immediately after or even during transcription and can be rapidly degraded by the cellular machinery [3,4,5]. These characteristics make lncRNAs sharp regulators of the myriad biological processes in which they are involved, such as chromatin remodeling, protein scaffolding or gene expression regulation through direct binding to genomic enhancers [6,7].

The study of lncRNAs from an evolutionary perspective has been hindered by their lack of strong primary sequence conservation [8,9], their apparent lack of secondary structure conservation [10], and their massive genomic generation and decay rate [11]. The cephalochordate *Branchiostoma lanceolatum* represents the earliest branching chordate lineage and holds a genome that seems to have retained many of the features of the ancestral pre-duplicative vertebrate [12,13]. Searches of lncRNAs conserved between amphioxus and vertebrates based on sequence similarity have been unsuccessful [14,15], probably due to the long evolutionary distance that separates these lineages [16]. Recently, however, a strategy to identify conserved intergenic lncRNAs (lincRNAs) by means of syntenic analyses has been successfully attempted, but limited to closely related species [17,18].

Here, we have developed a pipeline called LincOFinder that finds conserved clusters of microsynteny between two distant organisms surrounding an intergenic lncRNA. Furthermore, we use this tool to study the conservation and evolution of the lincRNA repertoire in the chordate lineage, finding up to 16 lincRNAs putatively conserved between amphioxus and human. Finally, we further study the case of *Hotairm1*, assessing its developmental expression in amphioxus and *Xenopus* and showing that its inhibition during *X. tropicalis* development produces a severe headless phenotype, probably by disrupting the chromatin dynamics of the anterior Hox cluster. Overall, our work pioneers the use of syntenic searches to identify non-coding genes over long evolutionary distances and helps to further understand lincRNA evolution in the frame of the invertebrate-vertebrate transition.

## 2. Materials and Methods

### 2.1. Amphioxus and Human Coding and lincRNA Datasets

We used the intergenic and bidirectional fractions from the lncRNAs dataset provided by Marlétaz et al. [15] to obtain an amphioxus lincRNA fraction (1318 genes), and their protein-coding genes supported by orthology as the coding fraction (10,832 genes). The human coding genes were obtained from the Ensembl annotation of the Ch38.96 genome assembly [19]. Finally, the orthologous gene families described in Marlétaz et al. [15] were used to assess the amphioxus/human gene orthologies.

### 2.2. LincOFinder (lincRNA Orthology Finder)

LincOFinder (https://github.com/cherrera1990/LincOFinder) is a program designed to identify shared microsyntenic clusters surrounding lincRNAs between two species. These are named the “reference” (*Ref*) and the “interrogated” species (*Int*). In the first, nonautomated step, the genes of both species (only the coding genes for *Int*) need to be arranged according to their position within the corresponding chromosome or scaffold, then a virtual coordinate according to their position is stablished (e.g., the first gene in chromosome A will be chrA-1, the second chrA-2, etc.). Furthermore, the orthologies between the genes of both species are established, using sets of known orthologous families or with the help of programs like Orthofinder [20]. Once the data is properly formatted as indicated in the ReadMe.md of LincOFinder, each annotated lincRNA from *Ref* is used as a reference point. The three upstream and three downstream genes neighboring the lincRNA are selected, and the orthology coordinates from *Int* are parsed into a distance matrix (Figure 1). The reasoning behind selecting only the three upstream and three downstream genes is to try to be astringent enough to comply with the definition of microsynteny [21] but at the same time allowing insertions and deletions up to a certain degree and the discovering, in case they exist, of larger clusters. Only genes present in the same chromosome are taken into account for distance assessment, and comparisons between paralogs of the same *Ref* gene are avoided. Then, a UPGMA hierarchical clustering algorithm [22] is used to create viable distance clusters (the ones that comply with the previously stated restrictions), and the cluster with the minimum distance between two neighboring genes is selected, thus identifying possible microsynteny clusters. If several possible clusters are formed, then they are displayed separately. These microsynteny clusters should be further filtered by selecting only those that harbor adjacent genes. Finally, candidates are manually curated by looking for the presence of lincRNAs in the microsyntenic region of *Int* (the algorithm is blind to *Int* ncRNAs due to the possibility of missing unannotated syntenic lincRNAs that could be, for example, present in the form of ESTs). This step can be done using a genome browser such us UCSC [23] (Figure 1).

### 2.3. Xenopus Embryos and MO Injections

All experiments were performed in compliance with the relevant laws and institutional guidelines of the University of East Anglia. The research has been approved by the local ethical review committee according to UK Home Office under Project License PPL 70/8876. *X. tropicalis* females were primed 24 h before the eggs were required and induced 5 h prior the experiment both with Chorulon (human Chorionic Gonadotropin). *X. tropicalis* males were also primed with Chorulon. Eggs were naturally obtained and fertilized with a sperm solution in Leibovitz’s L-15 Medium supplemented with 10% calf serum and left at room temperature for 5 min. After that, embryos were immersed in 0.05× MMR (Marc’s Modified Ringer’s) (50 mM NaCl, 1 mM KCl, 0.5 mM MgCl_2_, 1 mM CaCl_2_, 2.5 mM HEPES pH 7.5) for 20 min at room temperature, then washed in 2% l-cysteine pH = 8 for 7 min and rinsed several times with 0.05× MMR. Embryos were incubated in a BSA (bovine serum albumin) coated Petri dish in 0.05× MMR at 26 °C.

Morpholino oligos (MO) were designed and provided by Gene Tools. Morpholino sequences are: Standard control CCTCTTACCTCAGTTACAATTTATA, Hotairm1 AATCACTATTTGCTCCTTACCGGGT. Microinjections were carried out in 3% Ficoll PM400 (Sigma St. Louis, MO, USA) at 2-cell stage in both cells, and then embryos were incubated at 26 °C. Once *Xenopus* embryos reached the appropriate stage, they were snap frozen in liquid nitrogen for RNA extraction or fixed for whole mount in situ hybridization using MEMFA (3.7% formaldehyde, 1× MEM salts), then washed in PBST (PBS, 0.1% Tween), dehydrated in a serial dilution of ethanol and kept at 4 °C or −20 °C.

### 2.4. Xenopus RNA Extraction, cDNA Synthesis, PCR and Quantitative PCR

RNA was extracted using High Pure RNA isolation kit (Roche Basel, Switzerland) and 1 μg of RNA was taken to synthesize cDNA using Maxima First Strand cDNA synthesis kit (Thermofisher Waltham, MA, USA). RT-PCR was performed using SYBR Green detection method. Primers were designed using Primer3plus; *gapdh* was used as a control housekeeper gene. Primer sequences are indicated in 5′–3′ direction.

*hoxa1*—F: AGAAGTTTGCCGGTCTCCTT, R: AAGCCATATTCCCCAGCTTT

*hoxa4*—F: CAGTATCCACCCCGAAAAGA, R: GGGTTCCCCTCCACTGTAAT

*hoxa5*—F: GTCAGTGCAACCCCAAATCT, R: TTTCCTTCTGGCCCTCCTAT

*hoxa6*—F: GGAAGTACAGCAGCCCTGTC, R: GTAGGTCTGCCTCCCTCTCC

*hoxa7*—F: GACTCCCATTTCCGCATCTA, R: GGTAACGGGTGTAGGTCTGG

*gapdh*—F: ACTACCGTCCATGCCTTCAC, R: TCAGGGATGACTTTCCCAAC

For RT-PCR, ~100 ng of cDNA was used for amplification and the total PCR product was loaded to a 10 mg/mL agarose gel.

P300—F: GATTGCTACACCACCTTCTC, R: CCATGGGAGTCTTGACAATC

*hotairm1*—F1: CACAGTGCAGATGTCAGTGC, F2: CTACGGAGAGATACTTGCAC, R1: ATGCACGGTGTGATCAGTCG, R2: AAGCAATAACCGAGGCCTCT

### 2.5. Xenopus Whole-Mount In Situ Hybridization

In situ hybridization was carried out as previously described [24]. Probes were synthetized for *X. tropicalis hoxa1* and *hotairm1* using the following primers (sequences 5′–3′) hoxa1F: GATCGTTTTGTGGTCGGACG, hoxa1R: GCAGCAATTTCTACCCTGCG, hotairm1F: CTACGGAGAGATACTTGCAC, hotairm1R: AAGCAATAACCGAGGCCTCT.

Otx2 and engrailed vectors for probe synthesis were kindly provided by Professor N. Papalopulu (Manchester).

### 2.6. Amphioxus Embryo Collection and Whole-Mount In Situ Hybridization

Ripe adult amphioxus specimens were collected in Banyuls-sur-mer, France. Spawning was induced as previously described [25] in a dry laboratory. After in vitro fertilization, embryos were cultured at 18 °C until they reach the desired stage and fixed with 4% PFA in MOPS buffer overnight at 4 °C.

The hybridization chain reaction (HCR) [26] in situ v3.0 kit by Molecular Instruments (Los Angeles, CA, USA) was used following the protocol provided by the manufacturer for zebrafish embryos, with some adjustments to the probe and hairpins concentration (2pMol and 18pMol respectively) and using nests with a 0.4 µm mesh. The sequence provided for the probe synthesis for *Hotairm1* was the following (5′–3′) AAGGAGAGACGAAAGTCACCGGGACAAACCGGAGGATGTCTCGGAGGACCCTACCACCGCTCCCGCCTGTGCTCTACAGGTCACCAGGTGGGGATAGCACAACATGTCCTCTCTAGACATCTCTACTACACGCAGCTTGCTACCTGAAAGTTATCATATCTAGAATGTATATCTGCTTCAGTGTAAGCAACG.

## 3. Results and Discussion

### 3.1. Conserved Microsynteny Clusters

In order to identify conserved lincRNAs across chordates, we developed a pipeline called LincOFinder and used previously described *Branchiostoma lanceolatum*-*Homo sapiens* orthology families to detect conserved microsynteny clusters around specific amphioxus lincRNAs [15,23] (see Methods). Here we present the most reliable set of homologous lincRNAs that we were able to produce. Although in our pipeline three upstream and three downstream genes are considered, the output must be trimmed to extract the bona fide orthologous lincRNAs. In this case, we decided to restrict the distance between coding genes to one, and to consider only the clusters formed by one upstream gene, the lincRNA and one downstream gene. From the 32 clusters, only the 16 presented in Table 1 were considered to have a bona fide orthologous lincRNA. We also analyzed under these restrictions the clusters formed by two upstream genes and the lincRNA and by the lincRNA and two downstream genes (Appendix A). The rate of orthologous lincRNA finding was around 45% in the aforementioned analyses and the whole raw output is available in the Appendix A. Using this approach, we were able to obtain a list of 16 lincRNAs putatively conserved between human and amphioxus (Table 1). To our knowledge, this list represents the best set of highly curated lincRNAs with the deepest evolutionary conservation reported to date [27]. The main advantage of LincOFinder over other methods based on lincRNA sequence conservation is that it relies on microsyntenic conservation and a proper establishment of interspecific orthology relationships, which are more evolutionary constrained than the highly mutable nucleotide sequences of lincRNAs. In conclusion, LincOFinder can help to uncover conserved lincRNAs over deep evolutionary distances, in any species for which proper gene annotation data is available.

### 3.2. Conservation of HOTAIRM1 across Chordates

*HOTAIRM1* was selected for further study because its conservation across several vertebrate lineages has been previously underscored [28,29], and its mechanism of action has been thoroughly studied [30]. *HOTAIRM1* was identified for the first time in myelopoietic human cells [31], during a screen for transcriptionally active intergenic elements within the HoxA cluster. In amphioxus it is situated in the Hox cluster between *Hox1* and *Hox2*, and between *Hoxa1* and *Hoxa2* in vertebrates (Figure 2). According to our microsynteny-based analysis, *Hotairm1* is conserved in most of the chordate species analyzed, with the notable exception of zebrafish (Figure 2). Nonetheless, *Hotairm1* appears to be present in other actinopterigians like spotted gar (data not shown), as well as in teleosts like medaka (Figure 2). Given that the Hox cluster is disintegrated in tunicates [32] and due to the absence of an antisense transcript 5′ of *Ciona intestinallis Hoxa1*, we were unable to confirm the presence of *Hotairm1* in this chordate subphylum. Finally, we could not find any trace of this lincRNA in the genomes of the cyclostomes *Eptapretus burger* and *Petromyzon marinus* possibly because the microsynteny is lost in this region, due to a lineage-specific loss or alternatively because the lncRNA annotation was deficient [33].

Our results add up to previous studies that have described the presence of *Hotairm1* in mammals [28], birds and reptiles [29], strongly suggesting that *Hotairm1* was retained within the HoxA cluster after the vertebrate-specific rounds of genome duplication [34], and that its origin predates, at least, the appearance of extant chordate lineages more than 500 million years ago.

### 3.3. HOTAIRM1 Expression Patterns in Amphioxus and Xenopus

The expression domain of *Hotairm1* is mostly unknown, although it has been observed to be significantly increased or decreased in several types of cancer [35,36,37]. Furthermore, its expression is dynamically regulated during neuronal differentiation, showing a sharp increase in early differentiating neurons [38]. According to available RNA-seq data [15], the expression of *Hotairm1* during *B. lanceolatum* development peaks at 27 h post fertilization (hpf) (Appendix A). To investigate *Hotairm1* expression during amphioxus development, we performed fluorescent in situ hybridization in embryos from 18 hpf to 48 hpf. At 18 hpf we couldn’t detect any signal, while at 21 hpf the expression of Hotairm1 appeared in scattered cells in the presomitic mesoderm and in the neural plate partly overlapping *Hox1* expression domain (data not shown) [39]. At 30 hpf, 36 hpf and 48 hpf *Hotairm1* expression is restricted to the neural tube from the 5th somite towards the anterior developing neural tube, probably reaching the Di-Mesencephalic primordium (DiMes) [39,40] (Figure 3A’). Relevantly, the expression domain of *Hotairm1* in this developmental stage overlaps with *Hox1* which is also expressed in the developing neural tube and localized in the hindbrain (Figure 3C,C’) [41,42].

In the vertebrate *X. tropicali*s, the expression of *hotairm1* was detected in the midbrain, hindbrain and the pharyngeal arch (Figure 3B’). At the same time, *hoxa1* in this species is expressed in the pharyngeal arch and the anterior developing neural tube (Figure 3B) [43].

These results suggest that the expression domain of *hotairm1* is conserved between *B. lanceolatum* and *X. tropicalis,* being expressed in the anterior half of the developing neural tube and partly overlapping with *Hox1* and *hoxa1* expression, respectively (Figure 3).

### 3.4. HOTAIRM1 Function and Expression Conservation

Remarkably, *HOTAIRM1* has been described to act as a regulator of the chromatin state within the Hox cluster in human cells. Wang & Dostle [30] found that two *HOTAIRM1* isoforms, one spliced and one unspliced, play diverging roles in the regulation of the HoxA cluster chromatin state in presence of retinoic acid. Their findings indicate that the spliced isoform binds to the polycomb repressive complex 2 (PRC2) and changes the chromatin state repressing the medial HoxA genes (*HOXA4, HOXA5, HOXA6)*. The unspliced isoform, on the other hand, binds to the UTX/MLL complex and promotes the expression of the proximal HoxA genes (*HOXA1* and *HOXA2*).

To gain insight into the function of *hotairm1* we tried to alter the expression balance of its isoforms during *X. tropicalis* development. In order to achieve this, we used a morpholino oligonucleotide targeting the 3′ splice junction, thus forcing an isoform switch towards the unspliced state (Figure 4).

Strikingly, the morpholino treatment resulted in a headless tadpole-stage embryo (Figure 5A). This phenotype is characterized by the decrease of expression of brain markers such as *otx2* (forebrain-midbrain boundary marker) and *engrailed* (midbrain-hindbrain boundary marker) (Figure 5B,C). These results suggest that alterations in the balance of *Hotairm1* isoforms produce a severe disruption in the development of the anterior part of the central neural system (Figure 5).

Finally, in order to check whether the expression of HoxA genes was altered in MO-treated *Xenopus* embryos, we performed Real Time quantitative PCR at stage 18, when neurulation is taking place. Our results show a significant upregulation of medial Hox genes *hoxa5* and *hoxa6*, and a downregulation of *hoxa4*, compared with control embryos. Remarkably, no significant change in the expression of the proximal HoxA gene, *hoxa1*, was observed. These results suggest that *HOTAIRM1* function is partially conserved between *Xenopus* and human (Figure 6).

## 4. Conclusions

We have developed a novel pipeline called lincOFinder that establishes bona fide microsyntenic clusters to detect deeply conserved lincRNAs. Applying this tool to investigate the invertebrate-vertebrate transition, we have managed to identify 16 lincRNA putatively conserved between amphioxus and humans. To our knowledge, this represents the first successful identification of homologous lincRNAs over very long evolutionary distances. We show that one of these conserved lincRNAs, *Hotairm1*, is expressed along the anterior half of the neural tube during amphioxus and *Xenopus* development. The injection of MO targeting the 3′ splice junction triggers an imbalance between the spliced and unspliced form resulting in the disruption of the proximal and medial hoxa genes. This change in hoxa expression produces in a tadpole a patterning defect in the anterior neural system leading to a headless phenotype. However, further work needs to be done to elucidate the molecular mechanism underlying this severe phenotype. This nonetheless, is a reliable indicative that this lincRNA is at least to some degree conserved in amphioxus, *Xenopus* and human, allowing us to infer that it is conserved in the phylum Chordata and that regulation of the Hox cluster by lincRNAs may be traced back to the origin of chordates.

## Figures and Tables

**Figure 1 biology-08-00061-f001:**
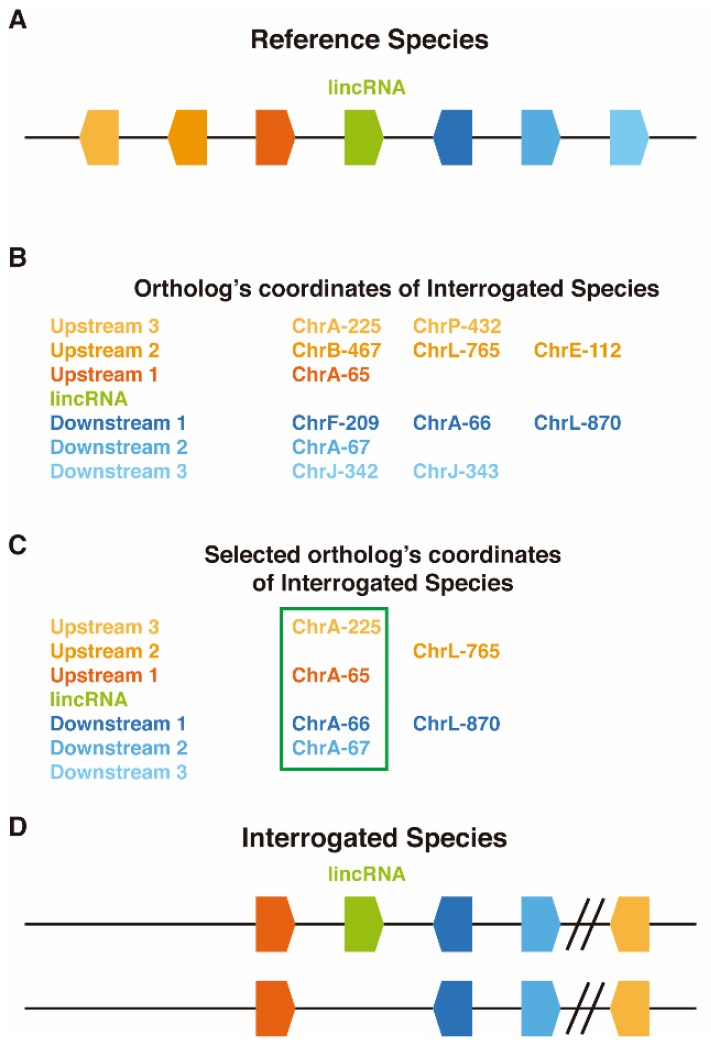
Diagram of LincOFinder mechanism. (**A**) Representation of the *Ref* species region where a lincRNA is present. (**B**) Formatted table of orthologs and virtual coordinates from the three upstream and downstream coding genes fed to the algorithm. (**C**) Selection of the best cluster according to the minimum distance between genes. (**D**) Representation of a conserved mycrosyntenic cluster in the *Int* species, where the presence of a lincRNA is manually confirmed (above) or discarded (below).

**Figure 2 biology-08-00061-f002:**
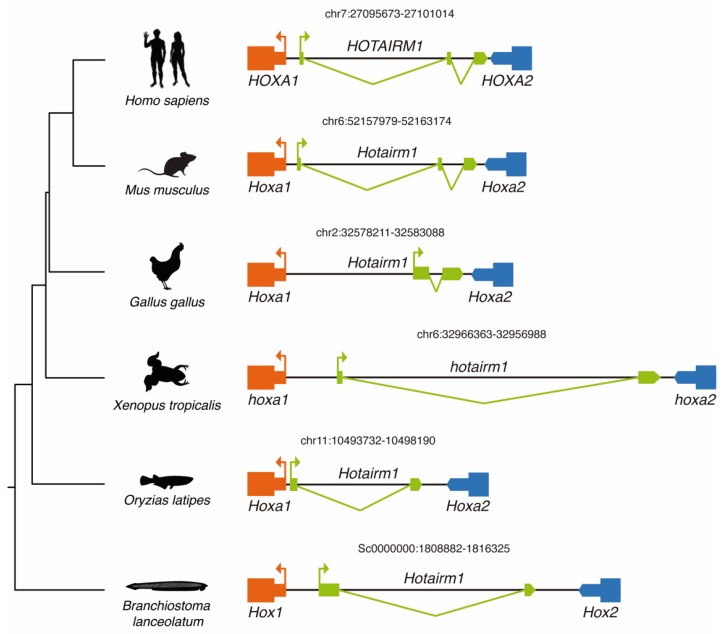
Schematic representation of the genomic locus of *Hotairm1* across several chordate species. Genome or scaffold position is indicated above each HotairM1 locus.

**Figure 3 biology-08-00061-f003:**
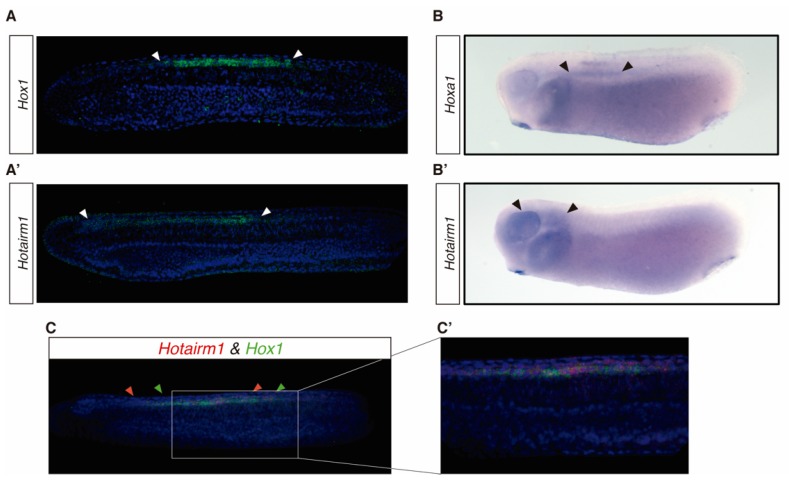
In situ hybridization (*ish*) in *B. lanceolatum* and *X. tropicallis*. Anterior to the left, dorsal is up. (**A,A’**) Fluorescent HCR *ish* in *B. lanceolatum* in whole mount for (**A**) *Hox1* and (**A’**) *Hotairm1* in 30 hpf embryos. White arrows mark the anterior and posterior limits of the expression domain. (**B,B’**) Colorimetric whole mount *ish* in *X. tropicalis* tadpoles for (**B**) *hoxa1* and (**B’**) *hotairm1*. Black arrows mark the anterior and posterior limits of the expression domain. (**C**,**C’**) Fluorescent double HCR *ish* in *B. lanceolatum* in whole mount *ish* of (**C**) *Hox1* and *Hotairm1* in a 36 hpf embryo and (**C’**) the detailed zone where *Hotairm1* peaks its expression. Green arrows mark the anterior and posterior limits of *Hox1* expression and red arrows mark the ones of *Hotairm1*.

**Figure 4 biology-08-00061-f004:**
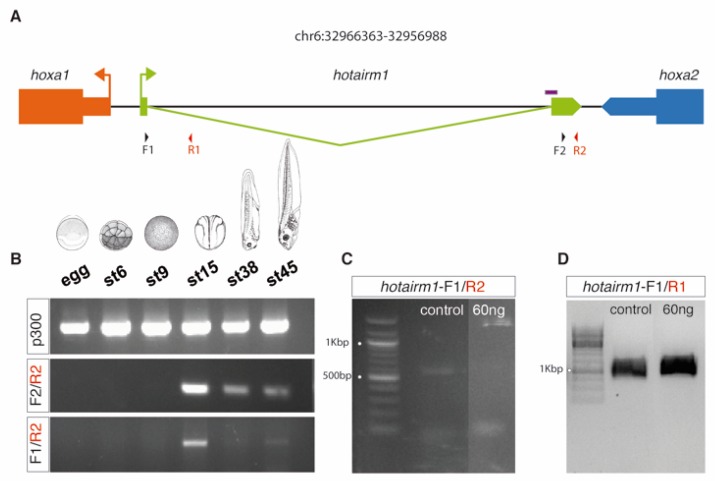
Isoform switch of *hotairm1* expression towards the unspliced state using a morpholino. (**A**) Detail of the primers (black and red arrowheads) and morpholino (purple box) used for the amplification of the spliced and unspliced isoforms of *hotairm1* in *X.tropicalis* and for the impairment of the splicing in the MO-treated embryos. (**B**) Expression of *hotairm1* across *Xenopus* developmental stages. (**C**) Inhibition of the spliced isoform in MO treated embryos of *Xenopus* at st18. (**D**) Assessment of the presence of the unspliced isoform of *hotairm1* in MO treated embryos as well as in the control embryos at st18.

**Figure 5 biology-08-00061-f005:**
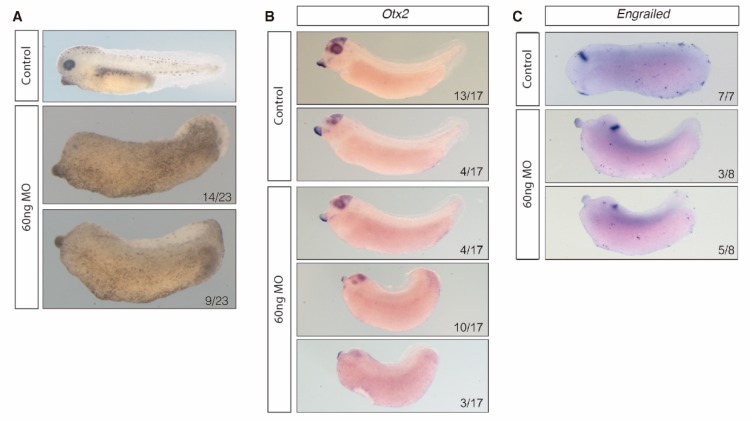
MO treated embryos and in situ hybridization in MO treated embryos. Anterior to the left, dorsal is up. (**A**) Control *X. tropicalis* MO treated embryos with normal development. 60ng hotairm1-MO treated embryos with a posteriorization of the anterior part of the embryo. (**B**) Whole mount colorimetric *ish* of *otx2* in *X. tropicalis* stage 26 control embryos and MO treated embryos showing the reduced expression domain of *otx2* in MO treated embryos. (**C**) Whole mount colorimetric *ish* of *engrailed* in *X. tropicalis* stage 26 control embryos and MO treated embryos showing a clear reduction in the expression in the MO treated embryos.

**Figure 6 biology-08-00061-f006:**
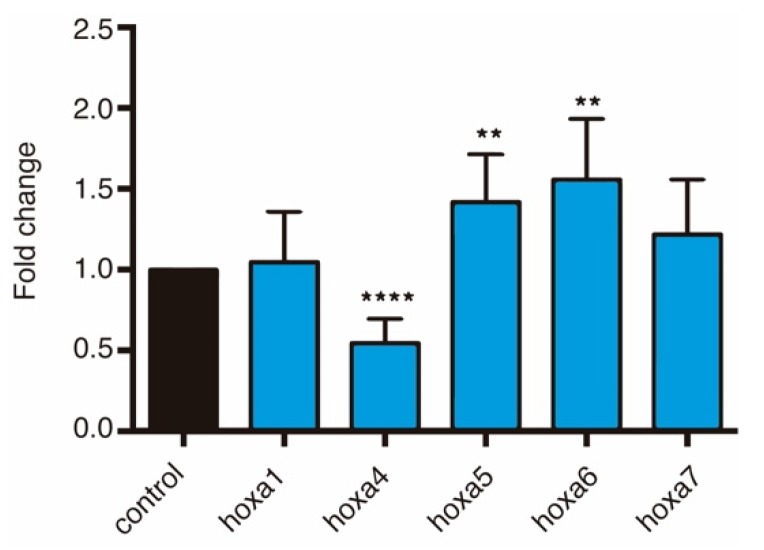
qPCRs between 60 ng MO treated embryos and control samples at stage 18. * shows statistically significance compared with control samples (Student’s *t*-test, * *p* < 0.05, ** *p* < 0.01, *** *p* < 0.001, **** *p* < 0.0001). *Gapdh* was used as a reference gene.

**Table 1 biology-08-00061-t001:** Putatively conserved lincRNAs between *Homo sapiens* and *Branchiostoma lanceolatum.* Analysis of the genes surrounding the lincRNA focusing on the three core genes (upstream1, lincRNA and downstream1). Some lincRNAs can be ascribed to more than one hypothetically conserved microsyntenic cluster.

Orthologous lincRNAs ^1^	State of the Cluster in Human ^2^	Human Orthologous lincRNA ^3^
BL20528|Sc0000000|28|+	* Conserved microsynteny	ENST00000623777.1_1
BL38782|Sc0000000|30|+	* Conserved microsynteny	HOTAIRM1
BL90848|Sc0000001|150|−	Correct order but strands inverted. In addition, there are two lncRNAs surrounding the cluster	AL354977.2
BL79733|Sc0000007|52|+	One gene with the strand inverted	BANCR
	One gene with the strand inverted	TCONS_12_00008513
BL84418|Sc0000009|86|+	* Conserved microsynteny	TCONS_00027655
BL91140|Sc0000010|144|−	Couple of lincRNAs in amphioxus, synteny conserved in the coding genes, but not in the lincRNA	TCONS_00006308
BL91143|Sc0000010|145|-	* Conserved microsynteny	RP11-181G12.4
BL53024|Sc0000015|55|+	One gene with the strand inverted	TCONS_00027115
BL82992|Sc0000016|15|−	One gene with the strand inverted	TCONS_00000550
BL78145|Sc0000039|54|+	* Conserved microsynteny	TCONS_00024711
BL55463|Sc0000050|45|+	* Conserved microsynteny	TCONS_00011710
BL68900|Sc0000072|2|+	Synteny conserved in the coding genes	AI219887
BL54861|Sc0000089|4|−	Problematic region with several lincRNAs and massive distances	AC109136.1
	Problematic region with several lincRNAs and massive distances	AC124852.1
BL41904|Sc0000219|3|−	* Conserved microsynteny	TCONS_00007813
BL72725|Sc0000229|14|+	* Conserved microsynteny	BC043517
	* Conserved microsynteny	LINC00114
BL59605|Sc0000234|6|−	* Conserved microsynteny	TCONS_00011870
BL38170|Sc0000240|5|+	* Conserved microsynteny	LOC100132215

^1^ GeneID, Scaffold, virtual coordinates and strand separated by “|”. ^2^ Description of the synteny of the cluster status in human. ^3^ ID of the putative human orthologous lincRNA. * indicates a perfect match in strand and order of the three core genes.

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
