# Peer review of "Microsyntenic Clusters Reveal Conservation of lncRNAs in Chordates Despite Absence of Sequence Conservation"

_biology, 2019, doi:10.3390/biology8030061_

Round 1
Reviewer 1 Report
I enjoyed reading the manuscript Microsyntenic clusters reveal conservation of lncRNAs in chordates despite absence of sequence conservation by Herrera-Ubeda and collaborators. The paper describes a new pipeline to detect lncRNAs using syntenic information, and performs the functional characterisation of one of them (Hotairm1). The analyses are sound, the manuscript is well-written, and the conclusions interesting. I would like to congratulate the authors for their work.
The main criticism is that the way is written it looks like two different articles were stuck together. The first part dealing with the pipeline could use some more elaboration beyond a list of lncRNAs, and provide more info about them (e.g., expression data) and their surrounding genes (e.g. functions). Importantly, the clustering algorithm used needs to be better described as well as the rate of false results. There is no clear link between the pipeline and the selection of Hotairm1, which is a well-known conserved gene and didn’t need the pipeline to be identified.
Other than that, I really liked the manuscript and I believe the main findings are interesting. I have some other comments and questions that I hope the authors can kindly address:
Line 73, please change “orthologic” by “orthologous”. Three upstream and three dowsntream conserved genes seems a quite stringent criteria. The manuscript should justify this value, and explore the impact of other values. Gene identification pipelines need an assessment of false positives vs false negatives. I guess that with such demanding parameters, many bona fide lncRNAs are discarded. Figure 1, while I understand this is just a toy example, the chromosomal coordinates should be explained (e.g., what is the 225 in “Chr1-225”?). Line 172, previous sections only dealt with comparisons 1 vs 1, between amphioxus and humans. Analyses of all vs all justifying the selection of Hotairm1 in Xenopus should be presented. Line 237, the proportion of embryos showing a headless phenotype should be stated Line 252, expression changes in MO-treated are well presented, but the conclusion about conservation of function compared to humans should be more elaborate.Author Response
Please see the attachment

Reviewer 2 Report
This paper introduces LincOFinder, a novel pipeline for lncRNA detection across long evolutionary distances, alongside an experimental characterisation of the expression of Hotairm1, a particularly strong candidate identified by the former.
The novelty of the approach has considerable potential to identify bona fide lncRNAs and so is likely to be of broad interest. However, LincOFinder is not publicly available – this is regrettable as by naming the pipeline and making it the first point in the abstract, the implication is that it will be accessible.
If the pipeline itself is not going to be published, some of the wording can be changed to more accurately reflect the principal focus of the manuscript (i.e., that microsyntenic searches are of practical value), rather than emphasise the pipeline as a stand-alone utility. For instance, line 49 “here, we developed a pipeline called LincOFinder that…” could just be “here, we identified conserved clusters of microsyteny…”
I appreciate that the pipeline contains several non-automated steps and best acts as a range-finding utility for subsequent manual curation, and that as a consequence the tool itself does not necessarily need to be available. Nevertheless, some parts of the methods section are written as if to help the reader make best use of LincOFinder, such as line 73, which suggests the use of Orthofinder (not otherwise required for the data used in the paper, which drew upon pre-existing orthology relationships).
More importantly, the critical part of the pipeline, on line 78, is not completely described. Can ‘clustering algorithm’ be expounded (how does this actually work)? The word ‘viable’ also implies some sort of cut-off to determine which clusters are output by the algorithm (that is, to distinguish viable from non-viable), but this is not clear.
How are distances between overlapping genes considered? Line 81 implies that they are excluded, as the clusters are filtered to consider only adjacent genes, but can this be made explicit? What of a cluster that contains both adjacent and overlapping genes? Is the entire cluster discarded, or only the overlapping genes within it?
The title of Figure 1B is misleading as the figure does not show coordinates, but (I believe) distances. Is “Chr1-225” supposed to mean “coordinates of 1-225 (if so, on what chromosome?)” or “distance of 225bp on chromosome 1” (i.e. between ‘upstream 3’ and ‘lncRNA’)? Figure 1C only makes sense in the context of the latter.
Line 81 refers to manual curation but presumably this has to be by sequence inspection? If so, do these orthologous lncRNAs indeed have low sequence similarity? If not, it suggests they could possibly have been identified without LncOFinder. Some confirmation that this is not the case would be valuable as there is a subjective dimension to whether putative lncRNAs within a cluster are ‘confirmed’ or ‘discarded’ (line 90). There is little detail as to what was specifically done, although table 1 suggests that clusters with ‘massive distances’ (can an example be given?) are a reason for exclusion. Line 150 is also phrased in such that a way that suggests conserved human/amphioxus lncRNAs could be identified from microsyntenic clusters alone – no mention of the interim step of having to manually scrutinise the intergenic sequence.
Minor comments.
Line 37. ‘myriad of’ should be ‘myriad’.
Line 42. At the first mention of amphioxus the scientific name, Branchiostoma lanceolatum, should also be given (the latter is otherwise first used on line 148).
Line 48. The identification of lncRNAs by means of synteny has also been successfully applied to ruminant genomes (sheep, goat, cattle), i.e. more divergent than nematodes but less so than Xenopus/amphioxus (PMCID: PMC5926538).
Line 65. Can it be clarified how many species were considered in the data obtained from Marletaz et al – this is important because Figure 2 is a schematic representation of some of the species used, but not all, with the results section referring to species not shown (such as Eptapretus burger and Petromyzon marinus on line 178).
Line 73. ‘orthologic’ should be ‘orthologous’.
Line 76. ‘matrix distance’ should be ‘distance matrix’.
Line 77. ‘within orthologs of the same Ref gene’ – do you mean ‘between paralogs’?
Line 88. ‘feed’ should be ‘fed’.
Line 155. ‘nucleotidic’ should be ‘nucleotide’.
Lines 156 and 260. For consistency with previous use, ‘lincOFinder’ should be ‘LincOFinder’.
Line 161. ‘adscribed’ should be ‘ascribed’.
Line 162. ‘clusters’ should be ‘cluster’.
Line 163. ‘GenID’ should be ‘GeneID’.
Line 163 states ‘# position’ but could this be clarified as ‘start position’? Full start-end coordinates would be preferable.
Line 186. ‘specie’ should be ‘species’.
Line 266. ‘Xenopus’ should be italicised.
Reviewer 3 Report
This is a very interesting case study. I suggest to publish as is. It would be useful for readers, however, to add the coordinates, exact genome versions, and possibly also the sequences of the lincRNAs used in this study in a supplemental table.
Round 2
Reviewer 1 Report
I am satisfied with the changes made by the authors, I will like to congratulate them for their work.